

# Effects of afforestation on water resource variations in the Inner Mongolian Plateau

Qiang Xiao[1],[*], Yang Xiao[2],[*], Ying Luo[1], Changsu Song[3] and Jiacheng Bi[4]

[1] Chongqing University of Arts and Sciences, Chongqing, China
[2] College of Biology and Environmental Sciences, Jishou University, Jishou, China
[3] State Key Laboratory of Urban and Regional Ecology, Research Center for Eco-Environmental Sciences, Chinese Academy of Sciences, Beijing, China
[4] State Key Laboratory of Simulation and Regulation of Water Cycle in River Basin, China Institute of Water Resources and Hydropower Research, Beijing, China
[*] These authors contributed equally to this work.

## ABSTRACT

Afforestation is a key approach used to effectively prevent ecosystem degradation, which in itself is a key reason for the obstruction of sustainable societal development. In order to suppress sand and dust storms as a result of ecological environmental degradation in North China, the Government of China has sanctioned the planting of a large number of trees in Inner Mongolia. However, water resources in the Inner Mongolian Plateau are insufficient to sustain this effort because such a large number of trees consume a large amount of water, which also significantly increases evapotranspiration. This study uses spatiotemporal trend analyses and abrupt change analyses to determine the effects of afforestation on water resource variations in the Inner Mongolian Plateau. Results show that even though water resources in Inner Mongolia fluctuate, this resource has generally exhibited a declining trend from 1980 to 2015, corresponding to the NDVI trend. On spatial-temporal scales, water resources decreased significantly in the eastern section of the plateau, especially in the Horqin District and the Hulunbuir Plateau. By contrast, water resources increased as a whole in the western section of the plateau (Alxa Plateau). Driving analysis results show that water resource variation is mainly due to the contribution of change in precipitation (positive effect), which accounted for 39.35% of total changes in water resources, followed by the evapotranspiration (negative effect). In other words, afforestation with the primary aim of improving ecosystem has effectually upset the water resource balance of Inner Mongolia Plateau.

# INTRODUCTION

Being part of the Mongolian Plateau located in Eurasia, Inner Mongolia, whose terrain is dominated by plateau and mountainous areas, is an important ecological barrier in North China (*Ouyang et al., 2018*). In recent years, drought and anthropogenic impacts in the region have intensified sand and dust storms (*Cordeiro et al., 2018*). As a result, these particulates have changed the composition of dust clouds, which have affected most

Corresponding author
Yang Xiao, yangxiao84@hotmail.com

of the world to the extent that floating dust from Inner Mongolia can reach the Pacific Coast of North America (*Cheng et al., 2006*). Some studies have reported that overgrazing in the research area has led to the wilting of grassland and the formation of deserts (*Kong et al., 2015*; *DeCastro-Arrazola et al., 2018*). In actual fact, the plateau is comprised of some of the most vulnerable ecosystems in the world due to its particular environmental structure, composition, and location. Accordingly, the question of how to develop a new ecological resource management model specific to the region has been of key concern for the Government of China.

Water resources refer to water resources that can be used or may be used. They are an important part of natural resources, have sufficient quantity and appropriate quality, and meet the specific needs of a certain place for a period of time. China is a country with severe drought and water shortage. It is one of the thirteen countries with the poorest per capita water resources in the world. Moreover, the spatial and temporal distribution of water resources is very uneven. To mitigate or reverse the environmental deterioration of the region while protecting its local water resources, the Government of China launched a massive tree planting program in the Inner Mongolian Plateau that from 1955 to 2015 has exceeded 36,000 km$^2$ (*Fan, Harris & Zhong, 2016*). In spite of the fact that vegetation cover on the Inner Mongolian Plateau has increased in recent years, the impact of ecological restoration policies in this region progress has been made in research (*Cao & Zhang, 2015*).

Many countries are currently facing issues related to increasing water scarcity (*Alton, 2018*). It is estimated that available water per capita in China is only approximately one-fourth of the world average, and water resource per capita in arid areas of Inner Mongolia is only 355 m$^3$. In recent years, a large number of people in Shanxi Province, Inner Mongolia, and parts of Beijing and Tianjin cities and Hebei Province in North China has suffered from the destruction of the ecological environment and the scarcity of water resources resulting from conditions of almost continuous drought. Given that the Inner Mongolian Plateau is the primary ecological barrier of Inner Mongolia and the Beijing-Tianjin-Hebei region, for those who rely on this barrier, the methods used by the Government of China to protect the ecological environment and water resources on the plateau will have a significant impact on the effectiveness of the ecological barrier of the region (*Miao et al., 2015*; *Escobar-Flores et al., 2018*). Many studies have reported that China's large-scale water resource protection plans and water transport projects will significantly affect the ecological environment and available water resources in the Inner Mongolian Plateau and the Beijing-Tianjin-Hebei region. Northeast Asia is also subject to similar potential ecological impacts resulting from these plans and projects.

Inner Mongolia and the Beijing-Tianjin-Hebei region are the territories most affected by the relationship among variations in water resources, sand and dust storms, and afforestation (*Tagesson et al., 2018*). A number of water resource shortage studies in North China reported that this large-scale afforestation program will severely affect the amount of available water resources in the region (*Cao et al., 2016*). In addition, the effects of climate change on the amount of precipitation and vegetation cover in the Inner Mongolian Plateau will aggravate the impact of sand and dust storms in this region. A regions' available water resource typically leads to water competition among various species within an ecosystem,

as well as competition between natural ecosystems and human requirements. Researchers have predicted that river systems in China's Mongolian Plateau will effectively dry up due to the overexploitation of water resources of some rivers (*Feng et al., 2017*). In certain areas of the Inner Mongolian Plateau, the excessive use of groundwater has led to a serious decline in groundwater levels, causing a series of ecological problems, including large-scale vegetation mortality, an increase in sand and dust storms, and overall environmental degradation. Furthermore, biodiversity in the western region of the Mongolian Plateau will decline as the climate becomes progressively dryer and desertified areas expand (*Yang et al., 2019a*).

Although we are dependent upon ecosystem resources, our understanding of ecosystem processes is very limited (*Wilson & Norman, 2018*). Therefore, we need to better understand such processes and integrate this knowledge into the management of the ecological environment as well as formulate better policies and plans to manage it. The purpose of this paper is as follows: 1. The most important science-based problems that require urgent study in the Inner Mongolian Plateau are how to manage relationships among water resources, sand and dust storms, and afforestation. 2. how to overall variation trends of water resources and vegetation on the Mongolian Plateau, Causes of water resource variations on the Mongolia Plateau 3. how to evaluate the impact of afforestation on water resource variation to promote the effective utilization of water resources and reduce the impact of sand and dust storms.

## MATERIALS AND METHODS

### Study area

Inner Mongolia is located in the arid and semi-arid climate region. The average annual temperature is between $-5\sim10\,°C$, and the average annual precipitation is between $35\sim530$ mm, Under the comprehensive influence of temperature and water conditions, vegetation zones mainly show the spatial differentiation characteristics in the near meridional direction. From east to west, they are successively forest, forest grassland belt, typical grassland belt, desert grassland belt and desert belt. The dominant land type in the study region is grassland (Fig. 1), which includes steppe and sparsely distributed grass, comprising of approximately 73.2% of the plateau, mostly distributed throughout the central and eastern sections. With the increase of population and the development of economy, the disturbance degree of human activities on the grassland gradually enhance, resulting in a certain degree of ecological degradation in this region. In recent years, the Government of China has implemented several ecological protection and construction programs to alleviate grassland degradation in Inner Mongolia (Fig. 2), however, these programs could also significantly affect the eco-hydrological process of grassland ecosystems. The ecological protection and construction planning programs mainly include the Three-North Shelter Forest Program, the conversion from farmland to forest and grassland, sandy land control, and ecological relocation projects.

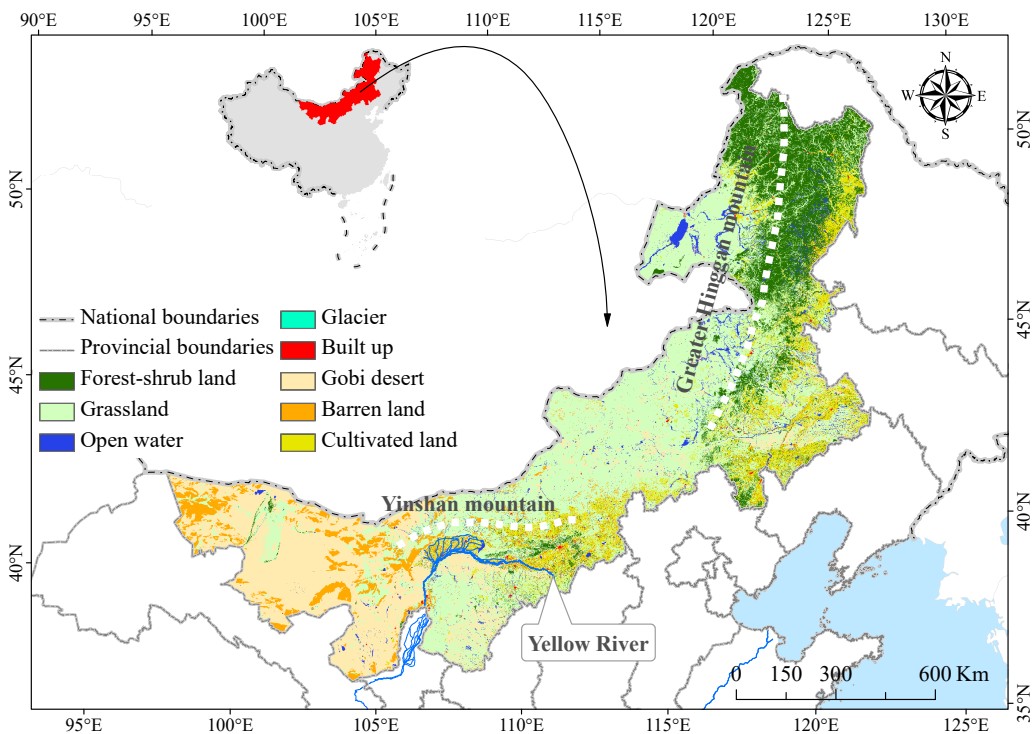

**Figure 1** Location and ecosystem distribution map of China's Inner Mongolian region.

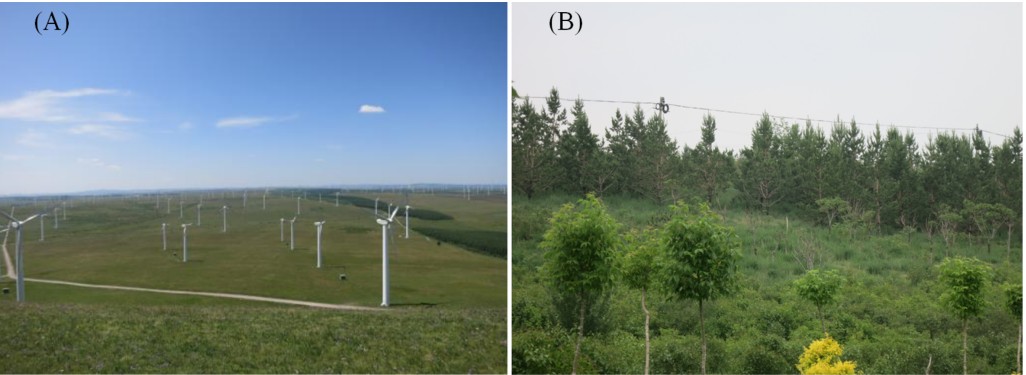

**Figure 2** Typical area characteristics of the research area. (A) Natural environmental characteristics of grassland ecosystem in the Tongliao region, where the wind is powerful enough to be suitable for wind power generation. (B) Local government planting water-intensive plantations in Hohhot to increase vegetation coverage.

## Data sources

In order to determine afforested area within the Inner Mongolian Plateau, we used afforestation data across Inner Mongolia from 1980 to 2015. We obtained afforestation data from annually published China forestry statistical yearbooks from 1980 to 2015 (State Forestry Agency of the People's Republic of China (1980–2015)). All cartographic data

were converted to the same coordinate system (Albers equal-area conic projection) and the same spatial resolution (1 km).

We obtained Chinese ecosystem data between 1980 and 2015 (the study period) from the Data Sharing Infrastructure of Earth System Science at the Chinese Academy of Science (http://www.geodata.cn). These data were produced at a resolution of 1 km, using visual interpretations of Landsat (MSS/TM/OLI) images with an average overall accuracy greater than 94%.

We obtained meteorological between 1980 and 2015 from the National Metrological Information Center of the China Meteorological Administration (http://data.cma.cn). Digital elevation model (DEM) data originated from the Shuttle Radar Topography Mission (SRTM), with a resolution of 90 m. Administrative divisions were provided by the Satellite Environment Center, the China Ministry of Environmental Protection (MEP). Lastly, we obtained the continental Global Inventory Modeling and Mapping Studies (GIMMS) normalized difference vegetation index (NDVI, NDVI = (NIR-R)/(NIR + R), NIR is near-infrared band, R is red band) dataset, with a spatial resolution of 8 km and covering a period from 1982 to 2015, from the Ecological Forecasting Lab at the NASA Ames Research Center (https://ecocast.arc.nasa.gov).

## Methods

(1) Water resource quantification:

Considering the special climatic and geographical environment in Inner Mongolia, there is a close interaction between climate factors, hydrocycle, and evaportranspiration. *Budyko (1974)* proposed a theoretical framework to describe the effect of precipitation and evapotranspiration on water storage based, which is an effective method to study the hydrological response to climate change. To quantitatively analyze water resources, we applied a water balance equation based on the Budyko model to the hydro series (from 1980 to 2015).

For this analysis, we do not consider the interaction between surface and groundwater on horizontal direction, due to the interaction usually happens in the vertical direction (*Xiao & Xiao, 2019*).

The water balance of a catchment can be described as follows (*Xiao & Xiao, 2019*):

$$Q = P - ET_a - \Delta S \qquad (1)$$

where $Q$ is the water storage (mm) as a proxy of the water resource; $P$ and $ET_a$ represent precipitation (mm) and actual evapotranspiration (mm); and $\Delta S$ represents a change in catchment water storage (mm), which is usually assumed to be zero over a long period of time (*Xiao & Ouyang, 2019*).

(2) Evapotranspiration factor *(ET)*:

Following an assumption similar to that made by *Budyko (1974)*, actual evapotranspiration can be estimated as follows (*Shen et al., 2015*):

$$ET = \frac{P \times ET_p}{(P^n + ET_p^n)^{1/n}} \qquad (2)$$

where $ET$ is the actual evapotranspiration (mm); $n$ is the model controlling parameter that determines the shape of the Budyko curve, which mainly represents the integrated effects of catchment land-cover characteristics on the water balance. $ET_p$ is the potential evapotranspiration, and it can be estimated by the Priestley–Taylor (PT) equation (*Priestley & Taylor, 1972*) as follows:

$$ET_p = \alpha \times \frac{\Delta}{\Delta + \gamma} \times (R_n - G) \tag{3}$$

$$\Delta = \frac{4{,}098 \times \left(0.6108 \exp\left(\frac{17.27T}{T+237.3}\right)\right)}{(T+237.3)^2} \tag{4}$$

$$\gamma = 0.665 \times 10^{-3} \times 101.3 \times \left(\frac{293 - 0.0065H}{293}\right)^{5.26} \tag{5}$$

where $\alpha$ is the PT coefficient of 1.26 for open water and saturated land (*Priestley & Taylor, 1972*); $\Delta$ is the slope of the saturated vapor pressure curve (kPa °C$^{-1}$); $\gamma$ is the psychrometric constant (kPa °C$^{-1}$); $R_n$ is the net radiation absorbed at the surface in MJ m$^{-2}$, which can be estimated using calibrated radiation in the FAO 56-PM model; $G$ is the downward-directed soil heat flux in MJ m$^{-2}$, where $G = 0.26 R_n$; $T$ is the mean air temperature (°C); $H$ is elevation above sea level (m).

(3) Spatiotemporal analyses:

The 'Spatial Analyst' tools in ArcGIS 10.3 were employed to reveal the spatial characteristics of water resources in different regions. To detect the variation in trends of water resources during the study period (1980–2015), a least squares linear regression model which is a commonly used method in trend analysis of variation, can be used to obtain the trend of every pixel change by fitting a linear equation of water resource variables as a function of time (year). The trend value was calculated by programming in Interactive Data Language (IDL) 4.8 (Harris Corporation, Melbourne, FL, USA).

(4) Abrupt change analyses:

We mainly conducted analysis of abrupt changes in water resources using the Mann-Kendall (MK) test (*Xiao & Ouyang, 2019*). We used this method on the premise that the positive sequence curve $UF_k$ crosses the critical ratio reliability line; therefore, if the positive sequence (UF) and the inverse sequence (UB, generated with the reverse data series of UF) have only one obvious crossing point located between the reliability lines, this denotes the catastrophe point and is statistically significant. On the other hand, if the crossing point is located outside the reliability line or there are several obvious crossing points between the lines, a definite catastrophe point cannot be established. In this latter case, we used the MK test on different lengths of sequences separately on the basis of a moving $t$-test technique. If a catastrophe point was still shown in these different sequences, we could then confirm that this point was the definite catastrophe point (*Pettitt, 1979*). The Mann–Kendall trend test can be described as follows:

For a time series $X = \{x1, x2, \ldots, xn\}$, the test statistic is given by

$$S_k = \sum_{i=1}^{i=k} n_i.$$

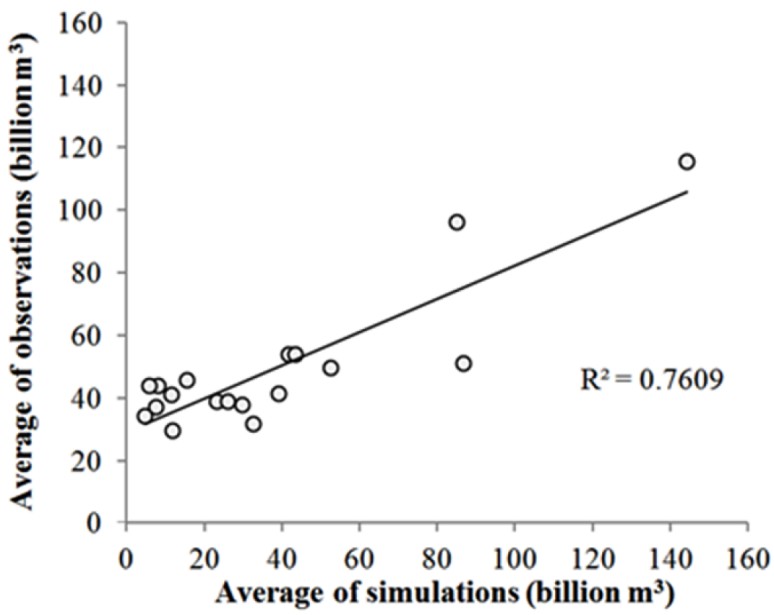

**Figure 3 Validation of water resource results from 1998 to 2015.**

The statistics time series $S_k$ is from $i = 1, 2, \ldots, k$, for any dual value $(x_i, x_j)$, if $j < i, x_i > x_j$, an upward trend is stated, $n_i$ is the total number of dual values with growth trend.

The expected variance of $S_k$ is calculated as follow,

$$E(S_k) = k(k-1)/4, \quad Var(S_k) = k(k-1)(2k+5)/72.$$

and the forward trend sequence $UF_k$ is given as:

$$UF_k = \frac{S_k - E(S_k)}{\sqrt{Var(S_k)}}.$$

## Validation

To verify the reliability of simulated water resources in Inner Mongolia, we validated simulated results against observed data, which are available from the China Water Resources Bulletin from 1998 to 2015 (Ministry of Water Resources of the People's Republic of China; 1998–2015).

# RESULTS

## Overall variation trends of water resources and vegetation on the Mongolian Plateau

Before the spatio-temporal analyses, we Validate the reliability of simulated water resources in Inner Mongolia, and the simulations agreed well with corresponding estimates (average

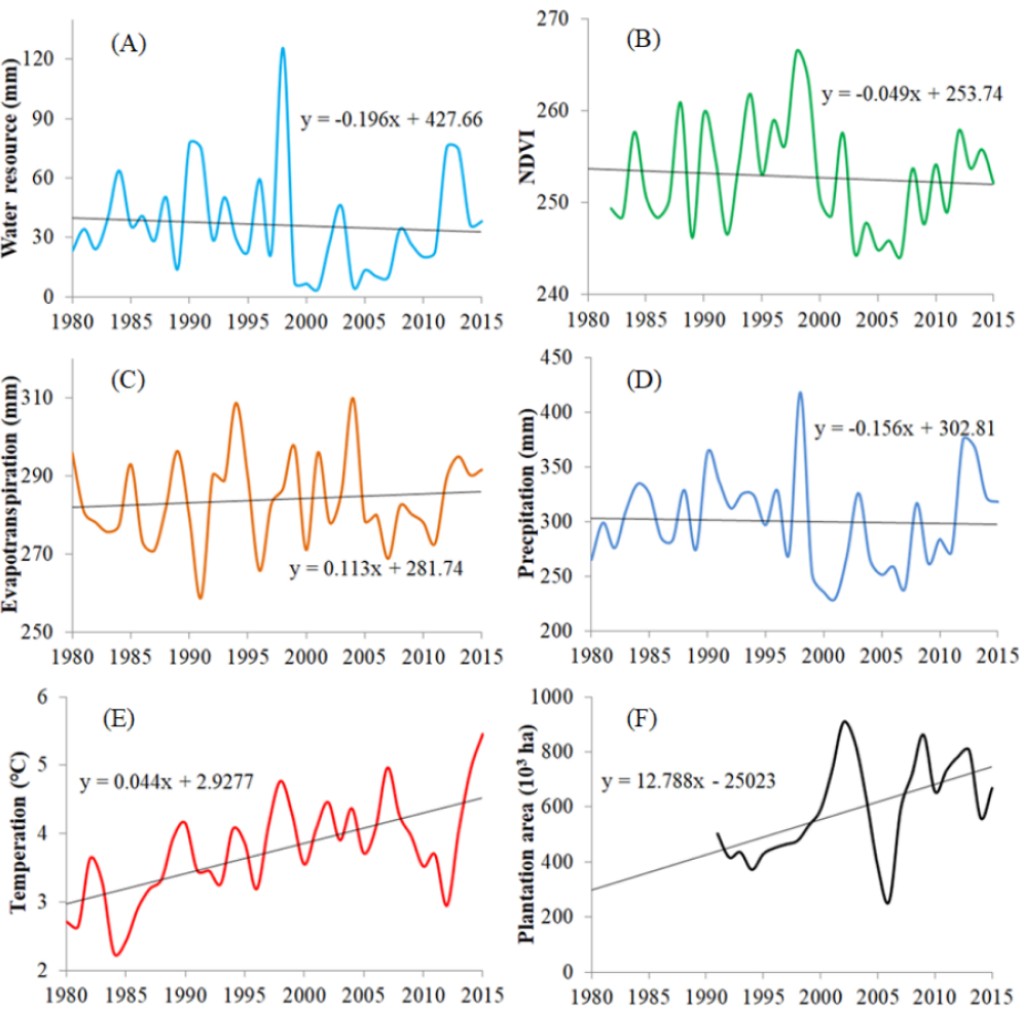

**Figure 4 Interannual variations of water resources (A), NDVI (B), evapotranspiration (C), precipitation (D), temperature (E), plantation area (F).**

$r = 0.84$; $p < 0.001$; Fig. 3). We could therefore confirm that simulated water resources could also be considered reasonable.

In recent years, water resources in Inner Mongolia have fluctuated but have generally exhibited a declining trend between 1980 and 2015 (a decrease of 0.2 mm/yr). This declining trend is unfavorable to vegetation growth. NDVI exhibited a similar trend. On a continental scale, 1998 generated the maximum water resource and NDVI values (Fig. 4). At the same time, Figure 4 also provides interannual variations and their influencing factors (such as temperature, evapotranspiration, precipitation, and plantation area) throughout 1980–2015. The decrease in rainfall and the increase in evapotranspiration in the region are not favorable to water resource storage. The characteristic of statistics of the six variables was shown as Table 1.

**Table 1  The description of statistics of the various driving factors and water resources from 1980 to 2015.**

|  | Mean | Maximum | Minimum | Standard error | Coefficient of variance |
|---|---|---|---|---|---|
| Water resource | 36.19 | 125.65 | 4.04 | 25.58 | 0.71 |
| NDVI | 299.93 | 417.83 | 229.68 | 42.13 | 0.14 |
| Evapotranspiration | 3.75 | 5.45 | 2.25 | 0.71 | 0.19 |
| Precpitation | 283.83 | 309.81 | 258.77 | 11.08 | 0.04 |
| Temperature | 252.79 | 266.43 | 244.21 | 5.73 | 0.02 |
| Afforestation | 591.67 | 907.39 | 256.52 | 170.56 | 0.29 |

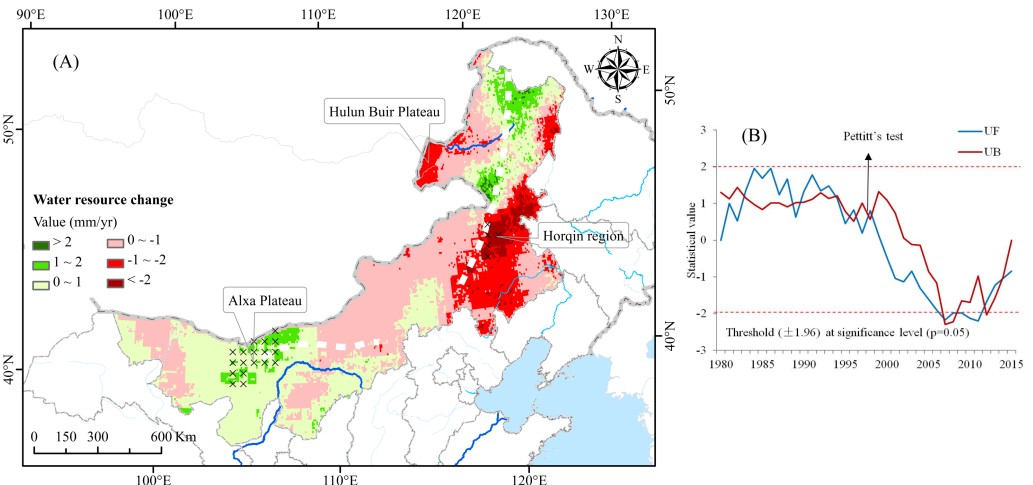

**Figure 5  Change trends and abrupt change tests of water resources in the Mongolian Plateau.**

## Declining water resources in the Mongolian Plateau

On a regional scale, almost the entire Mongolian Plateau has experienced water shortages between 1980 and 2015 (Fig. 5A). As a whole, water resources decreased significantly in the eastern section of the plateau, especially in the Horqin District and the Hulunbuir Plateau. By contrast, water resources increased as a whole in the western section of the plateau (Alxa Plateau). To evaluate long-term changes in water resources, we calculated the turning points of trends using the MK trend test and the Pettitt's test. As shown in Fig. 5B, we identified two distinctly different trends throughout 1980–1998 and 1999–2015.

## Causes of water resource variations on the Mongolia Plateau

There were obvious relationships among the various driving factors and water resources (Table 2). Our regression results found significant effects for six parameters associated with water resource changes from 1980 to 2015 (Table 3). For parameter value effects on water resources, the highest contribution was increasing precipitation (accounting for 39.35% of the total water resource changes), followed by increasing evapotranspiration (−24.48%), increasing DEM (22.72%), and increasing village density (−7.10).

**Table 2 The correlation analysis on relationship among the various driving factors and water resources.**

|  | Water resource | Precipitation | Temperature | Evapotranspiration | DEM | Slope | Village |
|---|---|---|---|---|---|---|---|
| Water resource | 1 |  |  |  |  |  |  |
| Precipitation | 0.78 | 1 |  |  |  |  |  |
| Temperature | 0.04 | 0.21 | 1 |  |  |  |  |
| Evapotranspiration | −0.49 | 0.01 | 0.31 | 1 |  |  |  |
| DEM | 0.45 | 0.57 | 0.50 | −0.03 | 1 |  |  |
| Slope | 0.09 | −0.23 | −0.34 | −0.50 | −0.06 | 1 |  |
| Village | −0.14 | −0.01 | 0.04 | 0.20 | −0.03 | 0.10 | 1 |

**Table 3 Regression results of relationships between the various driving factors and water resources after normalization. Contributions (%) represent the proportion of the total change in water resources accounted for by a given driving factor (Standard coefficient/Total absolute value of Standard coefficient).**

|  | $R^2$ | Adjusted $R^2$ | Contribution (%) | Standard coefficient |
|---|---|---|---|---|
| Precipitation | 0.61[**] | 0.609[**] | 39.35 | 0.781 |
| Temperature | 0.001 | −0.001 | 1.81 | 0.036 |
| Evapotranspiration | 0.236[**] | 0.235[**] | −24.48 | −0.486 |
| DEM | 0.204[**] | 0.202[**] | 22.72 | 0.451 |
| Slope | 0.008[*] | 0.006[*] | 4.53 | 0.090 |
| Village | 0.02[**] | 0.018[**] | −7.10 | −0.141 |

**Notes.**

$R^2$ is the regression goodness of fit based on the value of each driving factor. Parameters with significant contributions to regressions were identified by repeated measures ANOVA.

[*]$p < 0.05$.

[**]$p < 0.01$.

# DISCUSSION

## Temporal pattern of water resource variation

Results from our study clearly show that water resources in Inner Mongolia fluctuated significantly but generally exhibited a declining trend between 1980 and 2015 (a decrease of 0.2 mm/yr). NDVI exhibited a similar trend (*Miao et al., 2016*; *Shi et al., 2019*). These trends are not favorable to vegetation growth. Potential causes for these declining trends are that water consumption of artificial vegetation is far higher than that of natural vegetation; thus, the practice of restoring the environment and water resources in the Inner Mongolia Plateau by means of afforestation is likely to further aggravate existing water resource shortages (*Liu et al., 2018a*; *Cao et al., 2019*).

Figure 3 also shows interannual variations of influencing factors (such as temperature, evapotranspiration, precipitation, and planting area) from 1980 to 2015. Since 1978, a series of large-scale ecological restoration projects have been implemented in China, including the Three-North Shelter Forest Program, natural forest protection projects, and conversion projects from farmland to forest and grassland to name a few. Greater than 30 yr of practical experience has confirmed that green vegetation cover within the study area has rapidly increased by means of these environmental policies (*Lu et al., 2018*). However, compared to the natural recovery of abandoned land, afforestation in arid

areas has reduced total vegetation cover, increased degraded areas, and aggravated local desertification. The key internal mechanism that causes such water-related problems is as follows, when precipitation is less than evapotranspiration, vegetation (including trees) will be forced to rely on groundwater extraction to survive, leading to the depletion of surface water and a decline in groundwater levels, thereby reducing the availability of water resources for all vegetation (*Wu et al., 2017*). At the same time, a decrease in rainfall and an increase in evapotranspiration are not favorable to water resource storage.

Our results also showed that this water resource reversal in the Mongolian Plateau is intensifying, namely, that vegetation has begun to further degenerate due to large-scale afforestation initiatives (by an increase of $12.79 \times 10^3$ ha/yr) and global climate warming (by an increase of 0.044 °C/yr).

The MK trend test and Pettitt's test showed that there was an abrupt change in water resources around 1998, namely, water resources declined rapidly after 1998. This was mainly due to the fact that precipitation in Inner Mongolia has decreased yearly since 1998 while afforestation area has increased rapidly since that time, which has caused an annual increase in temperature and increased consumption of local water reserves. Along with this decline in water resources, the status of grassland vegetation has also gradually degenerated.

## Spatial pattern of water resources variation

Our research results clearly found a significant reduction in water resources in the eastern region of the Mongolian Plateau, especially in the Horqin District and the Hulunbuir Plateau. In contrast, there has been an increase in the total amount of water resources in the western region (Alxa Plateau) (*Yao et al., 2017*). The cause of this duel phenomenon is likely the implementation of ecological construction projects, such as natural forest resource protection initiatives (*Liu et al., 2018b*), the conversion from farmland to forest (*Chang et al., 2011*), the Three-North Shelter Forest Program (*Wang et al., 2007*), and compensation for public welfare forests (*Yang et al., 2012*), which have resulted in increased evapotranspiration and reduced rainfall in the region, while the usual practice of the Government of China is to simply promote a balance between different ecological service policies and practices.

With respect to the current increase in rainfall in the western region of Inner Mongolia, we believe this region should primarily be designated grassland. Namely, the protection and improvement of natural grassland ecosystems should be the key focus of the government, that it should abide by measures that combine protection and construction initiatives, and by sponsoring measures suitable to local conditions to further promote the ecological function of grassland and effectively alleviate grassland degradation and desertification trends, thereby developing a virtuous cycle (*Yang et al., 2019b*). We should also plan livestock feeding according to the availability of grass, promote a balance between grassland and livestock, convert grazing land to grassland, establish grassland protection areas, build artificial grassland areas in farming and pastoral farming areas, relocate population density beyond its environmental bearing capacity, etc., to strengthen the protection of grassland (*Fan et al., 2010*).

## Policy implications of water resources management

Being a public resource, the Government of China should instigate a comprehensive plan of action for water resources on a regional scale to ensure its sustainable utilization and to minimize negative impacts to the ecological environment. Similarly, prior to the implementation of an ecological policy, the Government of China must also take into account environmental factors, such as global climate change, sand and dust storms, as well as other relevant factors (*Chavas, 2017*). Concerning water resource variation on the Inner Mongolian Plateau, we found clear evidence that showed that water scarcity issues in the overall region primarily result from precipitation (*Wu et al., 2013*). Changes in precipitation have yielded a positive feedback on changes in water resources (a 39.35% contribution rate), while changes in water resources were primarily a consequence of increased evapotranspiration (a −24.48% contribution rate). A potential likely reason behind this is the large number of artificial forests that have been planted in the region in recent decades, resulting in increased evapotranspiration. Furthermore, village density in the Mongolian Plateau has had a negative impact on changes in water resources, which suggests that anthropogenic activities (such as the development of stockbreeding, afforestation, etc.) in recent years have had a significant impact on the region.

## CONCLUSIONS

Before we interfere with stable ecosystems, we must keep in mind that our behavior is constrained by ecosystem response mechanisms. This is because ecosystems are complex and are not yet fully understood. Any attempt to restore ecological environments should first be strictly accessed in order to expose possible unforeseen consequences. Whether the goal is to protect nature or to protect our own interests, it is vital that we be prudent in our actions. The results of this study show that no matter what measures are taken, the key point is to protect the dynamic balance among water resources, sand and dust storms, and afforestation in the Inner Mongolian Plateau. To achieve this goal, we need to adopt a holistic way of thinking. For example, we need to carefully judge and weigh the impact of every managerial action on water resources as well as the lives of people and ecosystems that are dependent on them.

Our results show the effects of afforestation on water resource variation in the Inner Mongolian Plateau. Over the last 30 years, vegetation evapotranspiration indices have risen, while large-scale afforestation has also resulted in an increase in total evapotranspiration (when compared to past evapotranspiration in the region), and subsequently the consumption of local water resources. Studies have reported that such degradation is due to the emphasis on short-term behavior and the neglect of ecological impacts. In this study, we determined the existence of a balance among some of the main ecological elements, which demonstrates that the best approach in protecting the ecological environment is to carefully maintain the original state of ecosystems.

## ACKNOWLEDGEMENTS

We would like to thank Denise Rennis for his help in writing this paper as well as journal editors and anonymous reviewers for their comments on an earlier version of this manuscript.

### Funding

The authors received no funding for this work.

### Competing Interests

The authors declare there are no competing interests.

### Author Contributions

- Qiang Xiao conceived and designed the experiments, contributed reagents/materials/-analysis tools, prepared figures and/or tables, approved the final draft.
- Yang Xiao conceived and designed the experiments, performed the experiments, analyzed the data, contributed reagents/materials/analysis tools, prepared figures and/or tables, authored or reviewed drafts of the paper, approved the final draft.
- Ying Luo performed the experiments, analyzed the data, contributed reagents/materials/analysis tools, authored or reviewed drafts of the paper, approved the final draft, translation.
- Changsu Song conceived and designed the experiments, performed the experiments, analyzed the data, prepared figures and/or tables, authored or reviewed drafts of the paper, approved the final draft, polished the article.
- Jiacheng Bi conceived and designed the experiments, performed the experiments, analyzed the data, contributed reagents/materials/analysis tools, prepared figures and/or tables, authored or reviewed drafts of the paper, approved the final draft, translation.

### Data Availability

The raw location and ecosystem distribution map of China's Inner Mongolian region is available in the Supplemental File.

### Supplemental Information

Supplemental information for this article can be found online at http://dx.doi.org/10.7717/peerj.7525#supplemental-information.

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
