# Peer review of "Effects of afforestation on water resource variations in the Inner Mongolian Plateau"

_PeerJ, doi:10.7717/peerj.7525_

## Round 0.1 · original submission · Major Revisions

Two reviewers have provided detailed comments on your manuscript with varying recommendations. Overall the topic of this study is important, however, both reviewers noted the obvious shortcomings. Reviewer #1 suggested to readdress the method and the conclusion, and directly point out the significance of this work. Review #2 questioned the novelty in this study, what's the different between this work and existing studies. The reviewers also questioned the use of certain approaches and interpretations of some results. Addressing the comments by providing more in-depth analysis, explanation of the methods used to detect the effects of afforestation on water resources, which are important for strengthening your study and improving the manuscript.

·

Basic reporting

Most of the English expressions are clear and unambiguous, professional.
Literature was well referenced and relevant, and sufficient field background/context were provided.
Article structure should be more clearly arranged.
The result part should be well modified.

Experimental design

The manuscript is original primary research within Aims and Scope of the journal.
Research question well defined, relevant & meaningful.
Methods described should be with more sufficient detail & information to make it more clearly so that it can be replicated just by reading the methods part in this article.

Validity of the findings

Data is robust, statistically sound, & controlled.
Conclusion are not well stated, and should be modified or rewritten to be linked to original research question and supporting results.

Additional comments

1. The language and format of this paper is fluent except for some few sentences, such as, line 35-37, line 84-86, sentences should be well arranged and some proprietary vocabularies should be corrected used.
please check the whole manuscript.
2. The abstract I think should be reorganized. Background sentences should be simplified or omitted. and the abstract should summarize the whole manuscript.
Here in the abstract, "This study uses GIS in order to determine the effects of forestation on water resource variations in the Inner Mongolian Plateau." I cannot find GIS method was described in the method part. I wonder what functions of GIS were used in the paper.
and "... this resource has generally exhibited a declining trend from 1980 to 2015, corresponding to the NDVI trend. " The resource decline? How? the quality or quantity?
the end part of the abstract should meet with the conclusion part of this manuscript.
3. Introduction
The introduction can be more logically narrated. Sufficient summery should be made to support the author's work, and in the end part of introduction, the work of this manuscript, maybe should be stated to show the significance.
4. figures should be referenced well in the text. see figure 1 and 2.
5. titles should not be followed by punctuations.
6. Line 141, Spatiotemporal analyses.
I cannot find Spatiotemporal analyses in the following statement. How Spatiotemporal analyses were carried out?
7. Line 164-167, should be placed in the results part.
8. Results.
More detailed analyses should be added in this part.
9. Discussion
Should be more closely related to the results of this study.
10. Conclusion
This part should be rewritten to conclude the main contribution of the authors. Some unrelated sentences at the beginning of the part could be omitted.

Reviewer 2 ·

Basic reporting

1. The literatures in the manuscript were insufficient to introduce the background of the water resources studies in the introduction section, as well as to explain the knowledge gap in the discussion sections.

2. The description of figures and table was also insufficient, more detail information is needed.

Experimental design

1. The manuscript need to more clearly define the research question and to explicitly indicate how the research fills the knowledge gap of water resources variation.

2. The description of method and materials were also insufficient.

Validity of the findings

1. Some of findings and decisions are not supported by enough literatures.

Additional comments

Exploring the effect of afforestation on the water resources is of great important to under the ecohydrological processes in semiarid environment. This study combined Bodyko model with GIS to indicate the distribution of water resources and their change processes in Inner Mongolian Plateau. The scientific question of this study is very important for the evaluation of ecological restoration policies. The results obtained by the study are meaningful, i.e., water resources decreased significantly in the eastern section of Inner Mongolian Plateau. I think these findings could attract many researchers to concentrate on the water resources in semiarid environment. However, a major revision needs to be performed before acceptation.

Annotated reviews are not available for download in order to protect the identity of reviewers who chose to remain anonymous.

---

## Round 0.2 · accepted · Accept

I am pleased to inform you that your paper has been accepted for publication. You were responsive to most of the comments made by the reviewers.

·

Basic reporting

Clear and unambiguous, professional English used throughout

Experimental design

Original primary research within Aims and Scope of the journal

Validity of the findings

All underlying data have been provided

Additional comments

The authors have answered all thequestions and made enough correction. I
think it can be accepted as is.

Reviewer 2 ·

Basic reporting

no comment

Experimental design

no comment

Validity of the findings

no comment

Additional comments

this revised article meets the PeeJ criteria